# Substance Use and Misuse among Adults with Chronic Obstructive Pulmonary Disease in the United States, 2015–2019: Prevalence, Association, and Moderation

**DOI:** 10.3390/ijerph19010408

**Published:** 2021-12-31

**Authors:** Abdullah M. M. Alanazi, Mohammed M. Alqahtani, Maher M. Alquaimi, Tareq F. Alotaibi, Saleh S. Algarni, Taha T. Ismaeil, Ahmad A. Alanazi, Moudi M. Alasmari, Eyas A. Alhuthail, Ali M Alasmari, Linda Gibson-Young, Wasantha P. Jayawardene

**Affiliations:** 1Department of Respiratory Therapy, College of Applied Medical Sciences, King Saud bin Abdulaziz University for Health Sciences, Riyadh 11481, Saudi Arabia; qahtani4@uab.edu (M.M.A.); alotaibita@ksau-hs.edu.sa (T.F.A.); qarnis@ksau-hs.edu.sa (S.S.A.); ismaeilt@ksau-hs.edu.sa (T.T.I.); 2King Abdullah International Medical Research Center, Riyadh 11481, Saudi Arabia; alanaiziah@ngha.med.sa; 3Department of Respiratory Care, College of Applied Medical Sciences, Imam Abdulrahman Bin Faisal University, Dammam 34212, Saudi Arabia; malquimi@gmail.com; 4Respiratory Services, Ministry of the National Guard—Health Affairs, Riyadh 11481, Saudi Arabia; 5College of Medicine, King Saud bin Abdulaziz University for Health Sciences, Jeddah 22384, Saudi Arabia; asmarim@ksau-hs.edu.sa; 6King Abdullah International Medical Research Center, Jeddah 22384, Saudi Arabia; 7Basic Sciences Department, College of Sciences and Health Professions, King Saud Bin Abdulaziz University for Health Sciences, Riyadh 11481, Saudi Arabia; EXA833@student.bham.ac.uk; 8Department of Biomedical Sciences, Institute of Clinical Sciences, University of Birmingham, Birmingham B15 2TT, UK; 9National Heart and Lung Institute, Imperial College London, London SW7 2BX, UK; a.alasmari18@imperial.ac.uk; 10College of Medical Rehabilitation Sciences, Taibah University, Medina 42353, Saudi Arabia; 11School of Nursing, Auburn University, Auburn, AL 36849, USA; gibsolm@auburn.edu; 12Institute for Research on Addictive Behavior, School of Public Health-Bloomington, Indiana University, Bloomington, IN 47405, USA; wajayawa@indiana.edu

**Keywords:** COPD, tobacco products, marijuana, substance use/misuse, serious psychological distress, gender

## Abstract

Background: Despite the advancements in chronic obstructive pulmonary disease (COPD) treatment, complications related to COPD exacerbation remain challenging. One associated factor is substance use/misuse among adults with COPD. Fewer studies, however, examined the prevalence and association between COPD and substance use and misuse. In addition, limited knowledge existed about the moderation effects of serious psychological distress and gender among adults with COPD and substance use/misuse. We aimed, therefore, to measure such prevalence, association, and moderation from nationally representative samples in the United States. Method: Data were drawn from the 2015–2019 National Survey on Drug Use and Health. Weighted logistic regressions were used to measure the associations of last-month tobacco (cigarettes, cigars, pipe, and smokeless tobacco products), other licit and illicit substance use (alcohol, marijuana, cocaine, crack, heroin, hallucinogens, and inhalants), and substance misuse (pain relievers, tranquilizers, stimulants, and sedatives) among adults with COPD. Serious psychological distress and gender were tested as moderators in the association between COPD and substance use/misuse. Results: The findings revealed that adults with COPD [_Weighted_ *N* = 53,115,718) revealed greater odds of cigarettes [adjusted odds ratio (aOR) = 2.48 (95%CI = 1.80–3.42)) and smokeless tobacco (aOR = 3.65 (95%CI = 1.75–7.65)). However, they were less likely to use alcohol (aOR = 0.61 (95%CI = 0.45–0.84)). Adults with COPD who had serious psychological distress were more likely to use pipe tobacco and alcohol; however, they were less likely to use hallucinogens and inhalants. Finally, males compared to females with COPD were less likely to use smokeless tobacco. Conclusion: Adults with COPD in the United States were more likely to use tobacco products and less likely to use alcohol. In addition, serious psychological distress and gender were moderators in associations between COPD and substance use but not in substance misuse. Future studies should longitudinally assess the factors that may contribute to the initiation and progression of substance use and misuse among adults with COPD.

## 1. Introduction

Chronic obstructive pulmonary disease (COPD) encompasses heterogeneous progressive respiratory disorders that include chronic bronchitis and emphysema [1]. Adults with COPD are presented with airflow limitation and frequent pulmonary complications [2]. Further, the pathophysiology of COPD reveals a variety of contributing factors in the lungs, including parenchymal destruction and narrowing of small airways diseases resulting from chronic inflammations [3,4]. Globally, the pooled prevalence of COPD was 13.1% where the prevalence of COPD was 6.0% in the United States [5,6]. COPD mortality in the United States has decreased from 2004 to 2018 (72.9 to 67.4 deaths per population of 100,000) [7]; however, significant diversity presents among adults with COPD based on mental health, gender, and risk-taking behaviors [7,8,9], one of which is smoking that has been documented as the most common risk factor for COPD etiology and exacerbation [9,10].

Smoking cessation is a standard of care for those who are diagnosed with COPD [11]; abstaining from tobacco use, indeed, showed promising health effects in limiting airway obstruction and ameliorating complications associated with COPD [12]. Despite the advancements in care, the associations between tobacco use and COPD remain high [13]. Tobacco use with COPD is not limited to tobacco cigarettes; there are, however, other forms of tobacco use that are observed among adults with COPD that include cigar, pipe, and smokeless tobacco [14,15,16]. Generally, the concern about substance use with COPD is rising to reduce the health care burden of the complications related to substance use among adults with COPD [17].

Previous studies concluded that substance use increased chronic disease complications, hospitalization, and mortality [18,19]. COPD, however, is not an exception, as alcohol, marijuana, and other licit and illicit substances have shown increased inflammatory markers of COPD [20,21,22]. Similarly, heroin, cocaine, opioids, and stimulants have manifested serious negative outcomes consequently with each use [22,23,24]. Nevertheless, little is known about the prevalence and association of substance use and adults with COPD in the United States.

COPD is frequently presented with unpleasant psychosensory symptoms of anxiety and pain [24,25]. It is unclear whether adults with COPD use non-prescribed substances such as painkillers and tranquilizers to alleviate such manifestations or for other off-label reasons [24,26,27]. Nevertheless, the side effects of misusing those substances could lead to catastrophic actions [27,28,29,30]. The extent to which adults with COPD use non-prescribed substances remains a concern with uncertainty [31].

Several explanations were proposed to explain the association between chronic diseases and substance use/misuse [32,33,34,35,36], one of which was seeking addictive substances as forms of negative affect avoidance and regulating mood disturbance that were associated with chronic disease manifestation [37,38]. An additional factor was the gender differences among those with COPD [36,39]. Previous research which examined the association between COPD and substance use/misuse accounted for gender differences as a covariate [40,41,42]; however, it is not clear if gender differences may moderate such association.

Smoking/smokeless tobacco products and substance use/misuse are pervasive among individuals with COPD, but little is known regarding the prevalence and association of substance use/misuse among the adults with COPD at the national level. This study aimed to: (1) evaluate the prevalence of substance use/misuse among adults with COPD; (2) assess the association between COPD and substance use/misuse; and (3) test the serious psychological distress and gender differences as moderators in the association between COPD and substance use/misuse. We hypothesized that the prevalence and association would be stronger among the individual with COPD. Furthermore, we hypothesized that serious psychological distress and gender differences would moderate this association.

## 2. Materials and Methods

### 2.1. Data Source and Sample

We used five waves of data (2015–2019) from the publicly available National Survey on Drug Use and Health (NSDUH) in the United States [43]. The NSDUH is a cross-sectional survey from a nationally and state representative sample of the civilian, non-institutional US population, ages 12 or older [43]. The original samples consisted of 282,768 respondents; after merging datasets, target participants younger than 18 years were excluded; 67,908 eligible respondents were included in the analysis.

### 2.2. Measures

#### 2.2.1. COPD Status

The NSDUH instructed the participants to list each health condition that they may have had during their lifetime. The directed query that assessed COPD “Have you ever been told you had COPD?” was selected. A dichotomous indicator of COPD status was created, with affirmative responses coded as “1” and negative responses as “0”.

#### 2.2.2. Substance Use

Last-month substance use was assessed from the NSDUH’s recorded drug use section for tobacco products and other licit and illicit substances. The participants were asked “How long has it been since you last smoked/used [the candidate substance being asked]?” The potential answers were as follows: 1 = within the past 30 days; 2 = more than 30 days ago but within the past 12 months; 3 = more than 12 months ago but within the past 3 years; 4 = more than 3 years ago; and 91 = never used. Any response that was not “1 = within the past 30 days” was treated as “0 = no.” The responses were then binary dichotomized (0 = no; 1 = yes).

The last-month tobacco use was assessed for the following tobacco products: cigarettes, cigars, pipe, and smokeless tobacco. Then, a global variable of “any tobacco use” was created (0 = no; 1 = yes) as an additional category. Any tobacco use refers to the users who consumed any tobacco products (cigarettes, cigars, pipe, and smokeless tobacco) in the last month.

Last-month substance use was assessed as well for the following licit and illicit substance groups: alcohol, marijuana, cocaine, crack, heroin, hallucinogens (LSD, PCP, Psilocybin/Mushrooms, Ecstasy, MDMA, Ketamine, Salvia Divinorum, or any other hallucinogens), and inhalants (not inhaled accidentally, but for kicks or get high). Further, a global variable of “any substance use” was created (0 = no; 1 = yes) as an additional category. Any substance use refers to the users who consumed any substance that was not tobacco (alcohol, marijuana, cocaine, crack, heroin, hallucinogens, and inhalants) in the last month.

#### 2.2.3. Substance Misuse

Last-month substance misuse was defined in NSDUH as “the use in any way not directed by a doctor, including drug diversion; use in greater amounts, more often, or longer than told to take a drug; or use in any other way not directed by a doctor.” Misuse of over-the-counter medications was not considered as substance misuse in NSDUH. Last-month substance misuse was assessed from the NSDUH’s recorded drug use section for the following drug groups: pain relievers, tranquilizers, stimulants, and sedatives. The participants were asked “In the past 30 days, did you use [the candidate substance being asked] in any way a doctor did not direct you to use?” The potential answers were as follows: 1 = yes; 2 = no; 91 = never used/misused pain relievers; and 93 = did not misuse pain relievers in the past 30 days. Every possible answer that was not (1 = yes) was treated as no (0 = no). The responses were then binary dichotomized (0 = no; 1 = yes). In addition, a global variable of “any substance misuse” was created (0 = no; 1 = yes) as an additional category. Any substance misuse refers to the users who consumed any non-prescribed medications (pain relievers, tranquilizers, stimulants, and sedatives) in the last month.

### 2.3. Covariates

#### 2.3.1. Sociodemographic Characteristics

Different sociodemographic characteristics were controlled in each model of substance use and misuse. Those covariates were as follows: gender (male, female); age (18–25 years, 26–34 years, 35–49 years, 50–64 years, or ≥65 years); educational attainment (high school or less, high school graduate, some college, associate degree, or college or higher graduate); race (White, Black, Hispanic, Asian, others, or multiracial); and income (<$20,000, $20,000–$49,999, $50,000–$74,999, or $75,000 or more). Sampling year (2015–2019) was controlled as a covariate as well.

#### 2.3.2. Serious Psychological Distress

Last-month serious psychological distress (0 = no; 1 = yes) was also considered as a covariate in each analytical model. Last-month serious psychological distress was measured in NSDUH by the validated Kessler Psychological Distress Scale [44].

#### 2.3.3. Co-Substance Use/Misuse

To account for the co-substance use/misuse with other substances, any substance use (0 = no; 1 = yes) and any substance misuse (0 = no; 1 = yes) were used in tobacco use models as covariates. Any tobacco product use (0 = no; 1 = yes) and any substance misuse (0 = no; 1 = yes) were used in substance use models as additional covariates. Finally, any tobacco product use (0 = no; 1 = yes) and any substance use (0 = no; 1 = yes) were used in substance misuse models as additional covariates.

### 2.4. Statistical Analysis

In the analytical procedure, COPD was introduced separately with moderators and covariates in each model as the independent variables. The dependent variables were each substance use/misuse. Each model was tested as the following example:
Last-month cigarette use = COPD × (COPD × serious psychological distress)× (COPD × gender) × covariates (serious psychological distress, gender, age, race,educational attainment, and income).

Frequencies were generated to assess the prevalence of last-month substance use/misuse among adults with COPD. Chi-square tests of independence were used for the comparisons between sample characteristics among adults with and without COPD. Bivariate analyses were generated using unadjusted logistic regression for the comparisons of all substance use/misuse in COPD versus non-COPD groups. Next, adjusted logistic regressions were used to assess the association between adults with COPD and substance use/misuse besides testing the moderation effects of last-month serious psychological distress (0 = no; 1 = yes) and gender differences (0 = females; 1 = males) in the association between COPD and substance use/misuse. The two moderators were introduced in the models that were “COPD (1 = yes) X serious psychological distress (1 = yes)” and COPD “(1= yes) X Gender (1 = males).” All analyses were conducted in STATA/SE 17.0 and accounted for the complex survey design weights by the NSDUH by the strata and clusters provided, as well as adjusting the analytical weights to account for five years (2015–2019).

## 3. Results

### Sample Characteristics

Sociodemographic data of the adults with COPD (_Weighted_ *N* = 53,115,718) and those without COPD (_Weighted_ *N* = 456,191,448) are shown in Table 1. In both groups, most respondents were females and whites. Older respondents, lower status of educational attainments, lower income, and greater serious psychological distress were more prevalent in the COPD group than those without COPD. Table 2 presents the prevalence of last-month substance use/misuse among both groups. Compared to those without COPD, adults with COPD reported significantly higher prevalence of cigarette and pipe tobacco products, except for smokeless tobacco. Moreover, they had significantly greater use of marijuana and less use of alcohol. Finally, adults with COPD reported significantly greater use of any tobacco products and any substance misuse; however, they reported less use of any substance that was not tobacco.

Table 3 depicts the odds of last-month substance use/misuse among adults with COPD. Accounting for covariates, those with COPD were more likely to use cigarettes (aOR = 2.48 (95%CI = 1.80–3.42)), smokeless tobacco (aOR = 3.65 (95%CI = 1.75–7.65)), and use any tobacco products (aOR = 2.57 (95%CI = 1.91–3.48)). However, they were less likely to use alcohol (aOR = 0.61 (95%CI = 0.45–0.84)) and use any substance that was not tobacco (aOR = 0.69 (95%CI = 0.51–0.93)).

To test the hypothesis that serious psychological distress and gender moderated the association between adults with COPD and substance use/misuse, these moderators were also reported in Table 3. Adults with COPD who have reported serious psychological distress in the last month were more likely to use pipe tobacco (aOR = 3.85 (95%CI = 1.46–10.13)) and alcohol (aOR = 1.64 (95%CI = 1.04–2.58)). However, they were less likely to use hallucinogens (aOR = 0.26 (95%CI = 0.08–0.81)) and inhalants (aOR = 0.14 (95%CI = 0.02–0.74)). Similarly, adults with COPD and reported their gender as males were less likely to use smokeless tobacco (aOR = 0.23 (95%CI = 0.10–0.50)) and any tobacco products (aOR = 0.54 (95%CI = 0.34–0.85)).

## 4. Discussion

This study assessed the prevalence and association of substance use/misuse among adults with COPD and whether serious psychological distress and gender differences moderated this association from nationally representative samples in the United States (2015–2019). The findings revealed that last-month use of tobacco cigarettes and smokeless tobacco were more likely among adults with COPD. However, those with COPD were less likely to use alcohol in the last month. Moreover, adults with COPD who had serious psychological distress in the last month were more likely to use pipe tobacco and alcohol in the last month. However, they were less likely to use hallucinogens and inhalants in the last month. Finally, male adults who have COPD were less likely to use smokeless tobacco in the last month.

The harmful impact of tobacco smoking is not limited merely to the onset of COPD [30] but accelerates the process of cell senescence, which decreases proliferation with preserved metabolic activity, contributing to increased inflammation, reduced cell regeneration, and carcinogenesis [45]. Similarly, smokeless tobacco product use poses an additional risk for COPD [16,46]. The dual use of smokeless and smoked tobacco is linked with an increased risk of cardiovascular disease, cancer, and mortality [16,47,48,49]. This suggests that smoking cessation should be capitalized on both forms of tobacco use among adults with COPD.

As a result of the chronicity and dependency of tobacco use, smoking cessation is extremely hard for those with COPD despite the health benefits of quitting [50,51]. The attachment to tobacco use is attributed to many factors, from social pressure to the inability to deal with triggers and cravings, and as an escape mechanism from negative affect [38,52,53,54]. A holistic smoking cessation approach should be emphasized in adults with COPD from pharmacological agents to behavioral support and psychosocial screening and prevention to optimize their treatment plans.

This study revealed, moreover, that adults with COPD were less likely to use alcohol in the last month. Alternatively, the study findings depicted those adults with COPD who self-reported serious psychological distress were more likely to use alcohol in the last month, which may explain that those who have COPD and experienced serious psychological distress may seek alternative mood regulator substances, such as alcohol [55]. Although alcohol use disorder is not directly associated with the onset of COPD, alcohol use disorder was associated with increased morbidity and health care utilizations among adults with COPD in the United States, which signified its impact with pulmonary diseases [56].

Generally, chronic health conditions were associated with a higher likelihood of any substance use and being more likely to be hospitalized for chronic health conditions [18]. Those with COPD in this study, however, reported significantly less use of any substance that was not tobacco than those without COPD [18,57]. It is warranted, therefore, to consider individual differences, such as serious psychological distress and gender, that may increase or decrease substance use/misuse among adults with COPD to optimize treatment and prevention services that ultimately improve health outcomes.

This study has some limitations. First, the sampling did not include homeless and active military people. Second, the design of the NSDUH is cross-sectional; hence, we cannot determine the temporality between COPD symptomology and substance use/misuse. Third, the clinical manifestations of COPD usually present at older ages; however, it was found in the NSDUH samples that some individuals who self-reported COPD were at younger ages (18–25 years), which threatened the accuracy of self-reported COPD in the samples. Finally, NSDUH samples did not ask about electronic cigarettes or vaping to assess new trends of emerging tobacco products among adults with COPD.

## 5. Conclusions

This study provided a better understanding of the prevalence, association, and moderation between adults with COPD and substance use/misuse in the United States. In general, adults with COPD were more likely to use tobacco cigarettes and smokeless tobacco. However, they were less likely to use alcohol. This study invokes the necessity of conducting longitudinal studies to determine the triggering factors and causal relationships more definitively between COPD diagnosis and substance use/misuse among adults to better design treatment intervention and prevention programs.

## Figures and Tables

**Table 1 ijerph-19-00408-t001:** Weighted sociodemographic characteristics of adults with COPD.

Variable	COPD = No	COPD = Yes	X^2^
_Weighted_ *N*	456,191,448	53,115,718	(*p* Value)
Gender			134.2 (*p* < 0.001)
Male	44.4%	36.8%	
Female	55.6%	63.2%	
Age			723.2 (*p* < 0.001)
18–25 years	20.7%	11.0%	
26–34 years	14.6%	10.0%	
35–49 years	26.6%	25.2%	
50–64 years	18.7%	25.0%	
≥65 years	19.3%	28.8%	
Educational attainment			490.2 (*p* < 0.001)
High school or less	11.9%	11.9%	
High school graduate	25.6%	31.3%	
Some college	24.0%	25.9%	
Associate degree	10.2%	9.5%	
College or higher graduate	28.3%	17.1%	
Race			391.7 (*p* < 0.001)
White	64.4%	74.7%	
Black	13.6%	9.3%	
Hispanic	13.1%	8.1%	
Asian	3.5%	1.2%	
Others	1.8%	1.9%	
Multiracial	3.6%	4.8%	
Income			587.8 (*p* < 0.001)
<$20,000	19.3%	28.7%	
$20,000–$49,999	30.9%	36.0%	
$50,000–$74,999	16.1%	14.5%	
≥$75,000	33.7%	20.8%	
Last-month serious			279.4 (*p* < 0.001)
psychological distress			
No	91.0%	84.5%	
Yes	9.0%	15.5%	

**Table 2 ijerph-19-00408-t002:** Weighted prevalence of last-month substance use and misuse among adults with COPD.

Substances	COPD = No	COPD = Yes	OR (95%CI)
Tobacco Cigarettes	31.2%	48.7%	2.08 (1.96–2.21)
Tobacco Cigars	14.5%	15.6%	1.08 (0.97–1.22)
Pipe Tobacco	7.0%	8.8%	1.28 (1.03–1.58)
Smokeless Tobacco	20.1%	16.3%	0.77 (0.66–0.90)
Alcohol	61.5%	49.2%	0.60 (0.57–0.64)
Marijuana	21.7%	23.2%	1.09 (1.01–1.18)
Cocaine	4.7%	4.0%	0.83 (0.63–1.10)
Crack	4.2%	4.5%	1.07 (0.69–1.68)
Heroin	9.0%	7.6%	0.82 (0.52–1.29)
Hallucinogens	3.1%	2.5%	0.79 (0.57–1.11)
Inhalants	2.2%	1.3%	0.62 (0.34–1.12)
Non-prescribed Pain Relievers	12.7%	13.7%	1.09 (0.90–1.31)
Non-prescribed Tranquilizers	15.6%	16.7%	1.08 (0.84–1.38)
Non-prescribed Stimulants	14.7%	13.1%	0.87 (0.64–1.19)
Non-prescribed Sedatives	7.5%	5.4%	0.71 (0.42–1.19)
Any tobacco use	39.8%	54.9%	1.84 (1.73–1.95)
Any substance use	64.1%	45.5%	0.67 (0.63–0.71)
Any substance misuse	21.7%	24.6%	1.17 (1.02–1.34)

**Table 3 ijerph-19-00408-t003:** The association between adults with COPD and last-month substance use/misuse.

Dependent Variables	Independent Variables		
	COPD	COPD X Serious Psychological Distress	COPD X Gender
Tobacco cigarettes	2.48 [1.80–3.42]	1.17 [0.73–1.88]	0.78 [0.49–1.24]
Tobacco cigars	1.29 [0.77–2.15]	0.95 [0.55–1.63]	0.83 [0.55–1.63]
Pipe tobacco	1.08 [0.36–3.22]	3.85 [1.46–10.13]	1.02 [0.37–2.80]
Smokeless tobacco	3.65 [1.75–7.65]	0.61 [0.27–1.39]	0.23 [0.10–0.50]
Alcohol	0.61 [0.45–0.84]	1.64 [1.04–2.58]	1.01 [0.62–1.60]
Marijuana	1.11 [0.81–1.51]	1.14 [0.72–1.79]	1.01 [0.65–1.55]
Cocaine	0.97 [0.44–2.13]	0.57 [0.18–1.83]	0.62 [0.31–1.22]
Crack	0.47 [0.13–1.62]	2.32 [0.49–10.91]	0.25 [0.04–1.33]
Heroin	0.69 [0.26–1.82]	1.01 [0.29–3.40]	1.08 [0.33–3.52]
Hallucinogens	0.98 [0.51–1.87]	0.26 [0.08–0.81]	1.66 [0.57–4.83]
Inhalants	1.28 [0.32–5.03]	0.14 [0.02–0.74]	1.33 [0.29–6.16]
Non-prescribed pain relievers	0.72 [0.42–1.24]	0.92 [0.44–1.92]	1.36 [0.78–2.38]
Non-prescribed tranquilizers	1.08 [0.55–2.11]	1.27 [0.57–2.85]	0.83 [0.37–1.83]
Non-prescribed stimulants	0.75 [0.33–1.70]	1.39 [0.58–3.30]	0.73 [0.31–1.71]
Non-prescribed sedatives	0.42 [0.15–1.15]	0.92 [0.20–4.24]	2.84 [0.59–13.49]
Any tobacco use	2.57 [1.91–3.48]	1.10 [0.70–1.71]	0.54 [0.34–0.85]
Any substance use	0.69 [0.51–0.93]	1.51 [0.90–2.52]	1.06 [0.66–1.69]
Any substance misuse	1.09 [0.77–1.55]	0.82 [0.52–1.28]	1.48 [0.94–2.35]

Adjusted Odds Ratios of last-month substance use/misuse among adults with COPD. All models were controlled for gender, age, educational attainment, race, income, last-month serious psychological distress, and sampling year.

## Data Availability

Publicly available datasets were analyzed in this study. This data can be found here: [https://www.samhsa.gov/data/data-we-collect/nsduh-national-survey-drug-use-and-health] (accessed on 9 November 2021).

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
