# Peer review of "Substance Use and Misuse among Adults with Chronic Obstructive Pulmonary Disease in the United States, 2015–2019: Prevalence, Association, and Moderation"

_ijerph, 2021, doi:10.3390/ijerph19010408_

Round 1

Reviewer 1 Report

This is a research paper aiming to understand the substance use/misuse among adults with chronic obstructive pulmonary disease (COPD) using the National Survey on Drug Use and Health (NSDUH).

This is an interesting study that could potentially raise concerns on substance use/misuse among COPD patients. However, the paper presented with poor analysis and little interpretation on the results. There are also minor issues in English writing, with the following suggestions that need to be improved or addressed:

  • According to CDC, health disparities are preventable differences in the burden of disease, injury, violence, or opportunities to achieve optimal health that are experienced by socially disadvantaged populations. Therefore, the title is misleading - smoking is the biggest risk factor for COPD, but doesn’t mean smokers have health disparity in getting COPD. Same with substance use/misuse. The study was looking at prevalence, not disparity.
  • It is not clear how data was analyzed in Table 1 – how the data was weighted, what variables were included.
  • Also not clear how the analyses were done in Table 2 - Chi square tests were used, which usually constitute contingency tables (2x2) and not a 1x2 table.
  • Results section was very short and not clear in many places – what covariates were accounted, how many patients were included in the analysis?
  • Figure 1: marijuana seems to be reduced the risk of COPD and not increased, and what about cocaine and tranquilizers?

Author Response

This is a research paper aiming to understand the substance use/misuse among adults with chronic obstructive pulmonary disease (COPD) using the National Survey on Drug Use and Health (NSDUH).

This is an interesting study that could potentially raise concerns on substance use/misuse among COPD patients. However, the paper presented with poor analysis and little interpretation on the results. There are also minor issues in English writing, with the following suggestions that need to be improved or addressed:

Response: Dear reviewer, on behalf of my co-authors, we would like to thank you for your time and input into our manuscript. Please, find below our responses and edits.

  • According to CDC, health disparities are preventable differences in the burden of disease, injury, violence, or opportunities to achieve optimal health that are experienced by socially disadvantaged populations. Therefore, the title is misleading - smoking is the biggest risk factor for COPD, but doesn’t mean smokers have health disparity in getting COPD. Same with substance use/misuse. The study was looking at prevalence, not disparity.

Response: We completely perceive your concern. Our intention was to address the disparity of substance use/misuse rather than the disparity of health. To avoid confusion, we have edited the title and manuscript and deleted the “disparity” term. Of note, to address the second reviewer’s comments, we have changed the title to be “Substance Use and Misuse among Adults with Chronic Obstructive Pulmonary Disease in the United States, 2015-2019: Prevalence, Association, and Moderation”

  • It is not clear how data was analyzed in Table 1 – how the data was weighted, what variables were included.

Response: We have added more description in the covariate section of the method about what we controlled for in the models. Plus, we have added more descriptions of the weighted process and variables used as recommended by the NSDUH.

  • Also not clear how the analyses were done in Table 2 - Chi square tests were used, which usually constitute contingency tables (2x2) and not a 1x2 table.

Response: Actually, table 2 represent 2X2 table because we used chi-square test between COPD (0=no, 1=yes) and substance (0=no, yes=1). In the table, we just reported only the percentage of use (1=yes) rather than both no and yes. Of note, to address the second reviewer’s comments, we have changed the analytical test in table 2 from chi-square to unadjusted logistic regression.

  • Results section was very short and not clear in many places – what covariates were accounted, how many patients were included in the analysis?

Response: To avoid redundancy, we highlighted most of the result findings in the narrative section of the results. Moreover, the weighted sample size was reported in the table and narrative section of the results. The controlled covariate in the models was described in detail in the method section and we edited it as well per your request as clarified in the above comment. Of note, to address the second reviewer’s comments, we have added more covariates that include “any tobacco use,” “any substance use,” and “any substance misuse.”

  • Figure 1: marijuana seems to be reduced the risk of COPD and not increased, and what about cocaine and tranquilizers?

Response: Because the figure was not exactly lining the ORs with its corresponding substance, marijuana was perceived as alcohol’s OR. We have changed the figure to table 3 to clarify each substance with its corresponding OR. Of note, to address the second reviewer’s comments, we have added two extra independent variables that were moderators (psychological distress and gender).

Reviewer 2 Report

Thank you for providing me the opportunity to review this manuscript, aimed to assess the relationships between COPD and substance use/misuse, among the 2015-2017 National Survey on Drug Use and Health (USA). Multiple substances were covered in the study: tobacco, alcohol, illegal drugs and other medicaments (like tranquilizers, sedatives, stimulants and pain relievers).  The study provides evidence regarding the different substance use-misuse among adults with COPD, and the associations between the presence of the disorder and the substances.

The objective of this study is really strong, and the manuscript could make an important and pertinent conceptual and empirical contribution to the scientific research. I believe that the paper will be candidate to be publishable, but major changes should be considered to make the study even stronger.

Please find below my considerations about each section of the manuscript.

Abstract:

- Define COPD in the abstract before using the abbreviation.

- The background in the abstract includes some not well-connected sentences. For example, the first sentence suggest that the study could be focused on the role of substances in the COPD treatment. Next, it is not clear the potential contribution of the study to the etiological area. And the third sentence suggest that a correlational analyses is performed.

Introduction:

- The first paragraph should be better centered on the definition of COPD (briefly, physiology and clinical profiles), epidemiology of COPD (prevalences in USA and other countries should be provided), and main risk factors.

- Paragraph-4 includes the use of non-prescribed medication as a potential risk factor for increasing the impairment of COPD. However, it seems that these pharmacological treatments are those used for alleviating the anxiety and pain symptoms related with the COPD. But individuals could use these substances for many other reasons, independent of the COPD. Please, clarify.

- The empirical hypothesis of the study is not well defined. I kindly suggest indicate multiple hypotheses in this line: a) the associations between substances and COPD will be higher for “indicate the substance” compared to “indicate substance”; b) compared with women, the association for men ….; c) considering the different groups of age, ….

Materials and Methods:

- Data analyzed in the study is obtained from the NSDUH, a public dataset. Authors provide a cite-reference [29], but I suggest a wider definition for the origin of the sample.

- Authors indicate that original sample consisted of 170,319 “unweighted respondents”, and that sample selected for the study consisted of 40,840 “unweighted elegible respondents”. Please, clarify the term “unweighted”.

- Please, clarify how tobacco and others substances were measured in the NSDUH’s.

- I do not understand the structure of the Materials subsections. For example, the first sentence in page 3 indicates “was regressed separately on COPD with covariates”. I consider that the Materials subsections should only indicate how the different variables analyzed in the study sere measured. Next, indicate in the statistical analysis section the analytical procedures: methods, independent/dependent variables of each model, covariate/s, …

- The covariates paragraph indicates that perceived overall health, and serious psychological distress were considered as adjust-controlled variables. Please, indicate how these variables were measured in the NSDUH.

- The covariates list did not include weight (or body mass index, or obesity). I suspect that this could act as a potential confounding variable.  For the same, other physical health conditions could be also acting as confounding factors. Why were not controlled these variables?

- Statistical analysis. Please, clarify how the weights were assigned in the study.

- Figure 1 is particularly interesting. I suggest including also the three global groups in the figure (tobacco, illegal drugs, non-prescribed medicaments).

- Table 2. I suggest change the chi-square and the p-value in the table with the coefficients obtained in the logistic regressions (B, SE, p, OR and 95%CI for OR). See the next comment regarding the adjusted-prevalences in the study.

- On the other hand, showing the results obtained, and due the high correlation between the use of different substances in the general populations, I consider that the analysis should be better carried out considering only two logistic regression models defining as the dependent variable the COPD (0=absent vs 1=present).  The first model should include in a first block-step the list of covariates, and in the second block the list of substances with (tobacco, alcohol, illegal drugs and non-prescribed medicaments). The second model should include in a first block-step the list of covariates, and in the second block the list of variables now plotted in Figure 1. These two models allows to assess the association between each substance adjusted to the use of the other substances.

- Finally, I consider that adjustment for sex and age is adequate. However,  It should be interesting also assess the potential role of these two variables. For this reason, I suggest obtain also the two global logistic regressions separately for men and women, and also separately for the groups of age of the study. Authors can also think about the analysis of other potential interactions.

Conclusions

- This section should be reviewed in the basis of the results obtained in the logistic regressions including simultaneously the list of substances, and the analysis of interaction factors.

Author Response

Thank you for providing me the opportunity to review this manuscript, aimed to assess the relationships between COPD and substance use/misuse, among the 2015-2017 National Survey on Drug Use and Health (USA). Multiple substances were covered in the study: tobacco, alcohol, illegal drugs and other medicaments (like tranquilizers, sedatives, stimulants and pain relievers).  The study provides evidence regarding the different substance use-misuse among adults with COPD, and the associations between the presence of the disorder and the substances.

The objective of this study is really strong, and the manuscript could make an important and pertinent conceptual and empirical contribution to the scientific research. I believe that the paper will be candidate to be publishable, but major changes should be considered to make the study even stronger.

Please find below my considerations about each section of the manuscript.

Response: Dear reviewer, on behalf of my co-authors, we would like to thank you for your time and input into our manuscript. Please, find below our responses and edits. In general, we have made the following changes:

  • To extra cycles of NSDUH to address the new moderation (interaction) variables in the study. The new sample consisted of 2015 to 2019 cycles.
  • Two moderators were created to account for the interactions with COPD. Those were psychological distress and gender differences. A full description of how we created them was reported in the method section.
  • We have edited the title, abstract, introduction, method, results, tables, and discussion to accommodate those changes.

Abstract:

- Define COPD in the abstract before using the abbreviation.

Response: It has been added.

- The background in the abstract includes some not well-connected sentences. For example, the first sentence suggest that the study could be focused on the role of substances in the COPD treatment. Next, it is not clear the potential contribution of the study to the etiological area. And the third sentence suggest that a correlational analyses is performed.

Response: We have edited the background section of the abstract to suit the study rational.

Introduction:

- The first paragraph should be better centered on the definition of COPD (briefly, physiology and clinical profiles), epidemiology of COPD (prevalences in USA and other countries should be provided), and main risk factors.

Response: The current structure of the first paragraph is centered on the definition with the clinical manifestation of COPD, epidemiology of prevalence, and mortality. Then we introduced the main risk factor (tobacco) along with the protentional moderators (psychological distress and gender).

- Paragraph-4 includes the use of non-prescribed medication as a potential risk factor for increasing the impairment of COPD. However, it seems that these pharmacological treatments are those used for alleviating the anxiety and pain symptoms related with the COPD. But individuals could use these substances for many other reasons, independent of the COPD. Please, clarify.

Response: We do agree that those who misused such medications may use it for other reasons that intended for. It is, unfortunately, unclear why those with COPD may use/misuse nor it is the scope of this study. We tried to introduce that we are not certain about extent of misusing such substances. However, we have added to this paragraph that those with COPD may misuse those medications/substances to alleviate pain and stress or for other off-label reasons. 

- The empirical hypothesis of the study is not well defined. I kindly suggest indicate multiple hypotheses in this line: a) the associations between substances and COPD will be higher for “indicate the substance” compared to “indicate substance”; b) compared with women, the association for men ….; c) considering the different groups of age, ….

Response: We have clarified the purpose and hypothesis of the study per your request.

Materials and Methods:

- Data analyzed in the study is obtained from the NSDUH, a public dataset. Authors provide a cite-reference [29], but I suggest a wider definition for the origin of the sample.

Response: We have edited the reference to get a wider definition of NSDUH

- Authors indicate that original sample consisted of 170,319 “unweighted respondents”, and that sample selected for the study consisted of 40,840 “unweighted elegible respondents”. Please, clarify the term “unweighted”.

Response: We have deleted the “unweighted” term to clearly describe that those numbers were for the number of respondents in the datasets. However, we kept the weighted number for the rest of the analysis since it was weighted.

- Please, clarify how tobacco and others substances were measured in the NSDUH’s.

Response: We have added additional clarification of the questions were asked and how each was coded in the NSDUH dataset and how we then recoded them in the analysis.

- I do not understand the structure of the Materials subsections. For example, the first sentence in page 3 indicates “was regressed separately on COPD with covariates”. I consider that the Materials subsections should only indicate how the different variables analyzed in the study sere measured. Next, indicate in the statistical analysis section the analytical procedures: methods, independent/dependent variables of each model, covariate/s, …

Response: We have made significant changes and addition to the material subsections. Especially, in the statistical analysis where we have added a clear description of how we carried out our analysis with an example of one model that was tested.

- The covariates paragraph indicates that perceived overall health, and serious psychological distress were considered as adjust-controlled variables. Please, indicate how these variables were measured in the NSDUH.

Response: After merging additional cycles and two account for the moderation analyses that we have added. We have limited the covariates to the sociodemographic only as a similar paper done with the NSDUH datasets. Sociodemographic characteristics account for significant diversity/variation in substance use and misuse which made us to control for them as covariates. We have edited how we structured the covariates to sociodemographic characteristics, psychological distress, and co-substance use/misuse.

- The covariates list did not include weight (or body mass index, or obesity). I suspect that this could act as a potential confounding variable.  For the same, other physical health conditions could be also acting as confounding factors. Why were not controlled these variables?

Response: Similar to the previous answer, accounting for such many covariates will limit the answers for infrequent substances reported such as crack and hallucinogens. We have mimicked similar studies that accounted for sociodemographic characteristics.

- Statistical analysis. Please, clarify how the weights were assigned in the study.

Response: We have added additional clarification of the weighting procedure in the method section.

- Figure 1 is particularly interesting. I suggest including also the three global groups in the figure (tobacco, illegal drugs, non-prescribed medicaments).

Response: Although, we have changed figure 1 to table 3, we have added those variables as dependent variables. They were named as follows: 1. Any tobacco use; 2. Any substance use; and 3. any substance misuse. We have clarified that the second variable “any substance use” that was any substance use that was not tobacco. We made it as follow to include alcohol with the category and account for marijuana as well. For example, if we name the variable as “illegal drugs” then we cannot include alcohol and marijuana in this category.

- Table 2. I suggest change the chi-square and the p-value in the table with the coefficients obtained in the logistic regressions (B, SE, p, OR and 95%CI for OR). See the next comment regarding the adjusted-prevalences in the study.

Response: We did change the analysis to unadjusted logistic regression and reported OR and 95%CI in table 2.

- On the other hand, showing the results obtained, and due the high correlation between the use of different substances in the general populations, I consider that the analysis should be better carried out considering only two logistic regression models defining as the dependent variable the COPD (0=absent vs 1=present).  The first model should include in a first block-step the list of covariates, and in the second block the list of substances with (tobacco, alcohol, illegal drugs and non-prescribed medicaments). The second model should include in a first block-step the list of covariates, and in the second block the list of variables now plotted in Figure 1. These two models allow to assess the association between each substance adjusted to the use of the other substances.

Response: Thank you for this significant input. We made significant change to our analysis based on your recommendation. First, we have clarified our dependent, independent, moderators, and covariate variables in the method. Then, we included two additional cycles to be from 2015-2019. After that, we have added two potential moderators based on the literature (psychological distress and gender). In addition, we have added three global variables (1. Any tobacco use; 2. Any substance use; 3/ any substance misuse). All of them were explained in depth in the method and analysis sections. We again, thank you for such significant input in our analysis and methods.

- Finally, I consider that adjustment for sex and age is adequate. However,  It should be interesting also assess the potential role of these two variables. For this reason, I suggest obtain also the two global logistic regressions separately for men and women, and also separately for the groups of age of the study. Authors can also think about the analysis of other potential interactions.

Response: We tried to be consistent with the literature in reporting our covariates which made us stick to the sociodemographic characteristics. However, we have added two additional moderatos as explained above which were (psychological distress and gender). Both of them based on the literature may interact with COPD to predict substance use/misuse. We have decided to include gender as a moderator (interaction variable) because it would give more insight into the effect of gender in this association. However, we were not able to account for age as a moderator or a separate group because it was reported in the NSDUH dataset as a multicategorical variable which will complicate the moderation/regression analyses. We believe, however, accounting for age as a covariate will suffice the association. 

Conclusions

- This section should be reviewed in the basis of the results obtained in the logistic regressions including simultaneously the list of substances, and the analysis of interaction factors.

Response: We have edited the discussion and conclusion accordingly.

Finally, on behalf of my co-authors, we deeply appreciate the efforts and time you spent on our manuscript. As you have seen by now that the method and analyses were edited significantly. We believe that you made this study much robust and clearer to contribute to the literature. Thank you.

Round 2

Reviewer 2 Report

Thank you for providing me the opportunity to review again this manuscript. The authors have been extremely responsive to the all my suggestion. All comments have been addressed extremely thoroughly which further increased the manuscript's quality and comprehensibility. I have not additional suggestions; overall I would recommend the paper for publication.